TECHNICAL RELEASE

# Long-read assemblies reveal structural diversity in genomes of organelles – an example with *Acacia pycnantha*

Anna E. Syme[1,2,*], Todd G. B. McLay[1,3,4], Frank Udovicic[1], David J. Cantrill[1] and Daniel J. Murphy[1]

1 Royal Botanic Gardens Victoria, Birdwood Avenue, Melbourne 3004, Australia
2 Melbourne Bioinformatics, The University of Melbourne, Parkville 3010, Australia
3 School of BioSciences, The University of Melbourne, Parkville 3010, Australia
4 Centre for Australian National Biodiversity Research, CSIRO, GPO Box 1700, Canberra 2601, Australia

## ABSTRACT

Organelle genomes are typically represented as single, static, circular molecules. However, there is evidence that the chloroplast genome exists in two structural haplotypes and that the mitochondrial genome can display multiple circular, linear or branching forms. We sequenced and assembled chloroplast and mitochondrial genomes of the Golden Wattle, *Acacia pycnantha*, using long reads, iterative baiting to extract organelle-only reads, and several assembly algorithms to explore genomic structure. Using a *de novo* assembly approach agnostic to previous hypotheses about structure, we found that different assemblies revealed contrasting arrangements of genomic segments; a hypothesis supported by mapped reads spanning alternate paths.

**Subjects** Software and Workflows, Bioinformatics, Taxonomy

**Submitted:** 04 October 2021

\* Corresponding author. E-mail: anna.syme@unimelb.edu.au

Preprint submitted at https://www.biorxiv.org/content/10.1101/2020.12.22.423164v1.full

## STATEMENT OF NEED

Genomes from organelles such as chloroplasts and mitochondria have predominantly been sequenced by technologies producing read lengths of between 75 and 300 base pairs (bp). These are considered "short" reads in comparison with newer technologies that can routinely produce "long" reads of 10,000 bp and longer.

Although some of the earlier organelle genome assemblies were inferred from long-range PCR and associated strategies [1], most existing assemblies are based on short-read sequencing data, which was comparatively easier and cheaper to obtain.

Short-read assemblies are challenging because repeats longer than read length cannot be unambiguously placed within the genome assembly. This is evident in chloroplast genome assemblies, where the inverted repeats (IR) are typically assembled into a single contig, and then manually duplicated in the assembly result to recreate the circular structure. There are several reasons why this is not ideal; for example, variation in repeats may not be captured, and the IR boundaries are not always reconstructed accurately.

Short-read assemblies have additional challenges for mitochondrial genomes because of their larger size and structural complexity. Whereas the chloroplast genome (plastome) in land plants is typically ~160 kilobase pairs (kbp) and circular, the plant mitochondrial

genome (mitome) is ~800 kbp and probably exists in multiple dynamic structures. Although the mitome has traditionally been represented as a single circular structure, there is physical evidence of multiple shapes [2]. Long sequencing reads indicate several linear, branched, or smaller circular structures [3, 4] that may recombine at repeat regions [5].

Long sequencing reads may be able to span repetitive regions (depending on length) and should better capture their placement in assemblies, thereby revealing more of the structural complexity in both plastomes and mitomes. Presently, long reads generated using technologies such as Oxford Nanopore and Pacific Biosciences (PacBio) have a higher error rate than short reads from Illumina. Thus, reads from both can be combined when assembling genomes: long reads can reveal broad organelle genome structure, and short reads can correct errors. This hybrid approach has demonstrated improved accuracy in organelle genome assembly [6].

Recent work using these combined technologies has been highly successful in revealing new information about organelle genome structure. Long reads provide strong evidence that the plastome exists in two structural haplotypes in equal proportions across land plants [7], which supports certain theories of recombination. Gymnosperm mitome assemblies based on long reads reveal considerable complexity in mitome structure, and branching may be related to recombination processes [4].

Here, we used long (Oxford Nanopore) and short (Illumina) sequencing reads to assemble the plastome and mitome of *Acacia pycnantha* (Golden Wattle, Australia's floral emblem; NCBI:txid880440). The iconic, economically important *Acacia* genus has more than 1000 species, yet currently, long-read organelle assemblies are lacking. Data from nuclear ribosomal DNA and plastomes have provided a good basis for phylogenetic investigation [8]. Three plastomes and one mitome for the genus are available in RefSeq [9], and 94 additional partial assemblies (incomplete, with gaps in non-coding, repeat rich regions) in the National Center for Biotechnology Information (NCBI) database, many from Williams *et al.* [8]. These new long-read assemblies will complement and expand our knowledge of *Acacia* organelle structures.

To further facilitate exploration of genome structure and to limit the introduction of bias or errors, we assembled sequencing reads *de novo*, rather than mapping them to an existing assembly. Analysis steps are automated in reproducible scripts with freely available tools.

## IMPLEMENTATION

### Obtaining organelle reads

We used an iterative approach to extract organelle-only reads from full genomic read sets containing a mixture of nuclear, mitome and plastome reads. First, gene coding sequences from related *Acacia* species were used as baits to extract organelle Nanopore reads. These reads were assembled, and the assembly itself used as bait for repeat organelle read extraction from the full Nanopore read set. These reads were assembled, and this second assembly was used as bait to extract short Illumina reads from the full Illumina read set. The short reads were then used to polish the assembly (Figure 1). Additional assemblies were completed in different configurations as discussed in more detail below.

### Assembling organelle genomes

Short-read technologies have provided the dominant source of organelle sequencing read data for the past 10 years. Therefore, by necessity, assembly tools have also been based on

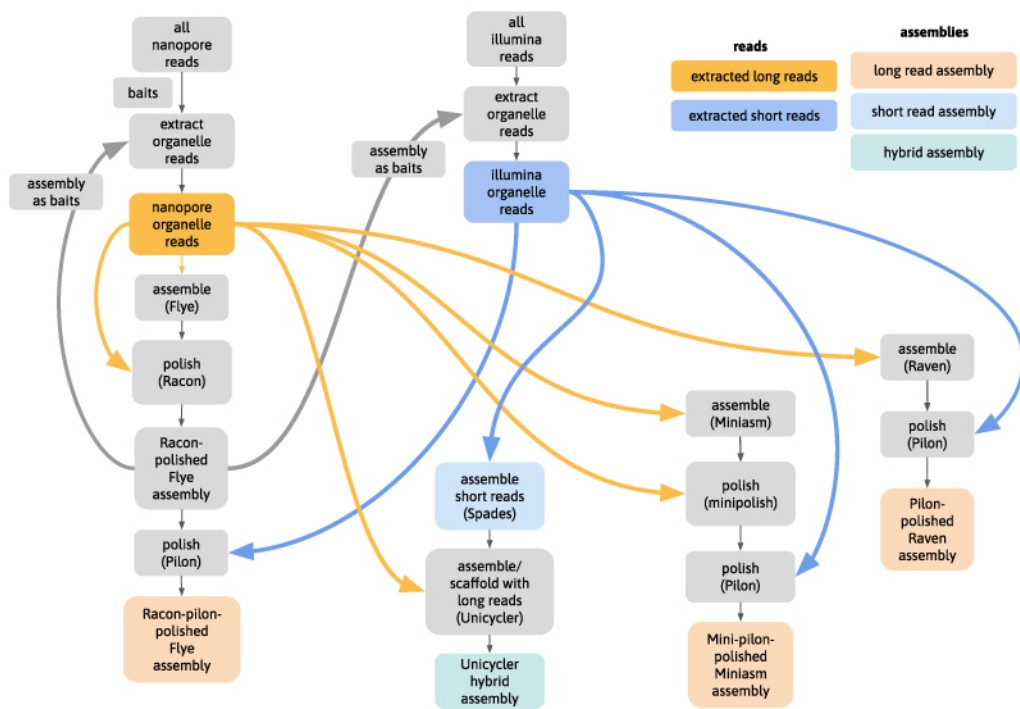

**Figure 1.** Analysis flowchart. Main steps in the analysis, showing use of initial assembly as baits for further read extraction, and use of read sets in assembly polishing steps.

reads of these lengths and fidelity, and have typically relied on mapping the reads to existing assemblies. More recently, the tools NOVOPlasty (RRID:SCR_017335) [1] and GetOrganelle [10] have been developed using iterative baiting algorithms to perform *de novo* organelle assemblies, and have improved accuracy. However, they are not configured to work with long-read data, which differs not only in read length, but in read length variability and a higher error profile [11]. Since we specifically wanted to investigate assemblies based on long reads, we therefore used assemblers optimized for these types of data.

## Choice of assembly tools

Organelles and bacteria have similar genomes because they probably have shared ancestry [12]. In assembling organelle genomes, it is therefore appropriate to consider methods used for bacterial genome assembly, which is an area of active research because accurate assemblies underpin many areas of public health. Recent benchmarking of bacterial assembly tools for long reads found the best performers to be Flye (RRID:SCR_017016) [13], Raven (RRID:SCR_001937) [14] and Miniasm [15] and Minipolish [16], but no single tool performed best on all metrics such as reliability, circularization, errors, or completeness [16].

Here, we used a combination of these well-tested assemblers, which also capture the diversity of algorithms in current use. For example, Flye uses approximate repeat graphs; Miniasm is a true Overlap–Layout–Consensus (OLC) method and only outputs unitigs; and Raven combines an OLC method with improved graph cleaning by removing unsupported



overlaps. All are designed to work with "noisy" long reads such as Oxford Nanopore sequences. Because we also include short reads in this analysis, we assembly by Unicycler [17] – a hybrid method using an initial short-read assembly followed by long-read scaffolding (Figure 1).

## Annotations

This analysis is primarily concerned with establishing first-pass assemblies for *Acacia* organelles, using long reads to explore structural configuration. Annotation is the next step to further investigate gene and feature content, arrangement, loss or duplication, transfer between organelles and nuclear genomes, and comparison with related and more distant species. In appreciation of the complexity of this, and the need for domain-specific knowledge to best produce a useful annotation, this work does not attempt to present a complete or final annotation of these organelle genomes. However, basic annotations are presented to provide an initial visualization of the feature landscape of these organelles. We used GeSeq [18, 19] to produce annotations, which is automated, reproducible, and has been used successfully for plant mitomes and plastomes [20, 21]. Future editing and refinements of these annotations are expected, and will no doubt improve the interpretation of structural and physiological processes. A continuous iterative process of refinement of assemblies, then annotations, then assemblies, and so on, would indeed be beneficial, particularly in the study of non-model organisms.

## Sample and sequencing

Young leaves were collected from a Golden Wattle, *Acacia pycnantha* (NCBI:txid880440), in the Australian National Botanic Gardens (voucher details: CANB 748486 S.R. Donaldson 3550 12/10/2007) and DNA was extracted from fresh tissue [22]. Sequencing was performed by the Australian Genome Research Facility (Melbourne, Australia). Oxford Nanopore sequencing used a PromethION R9.4.1 flow cell and basecalling with Guppy v.3.2.4, producing ~5.5 million reads (longest ~170 kbp, total ~60 gigabase pairs (Gbp)) (Table 1). Illumina sequencing used a TruSeq DNA Nano protocol, a NovaSeq 6000 S4 flow cell, basecalling with Illumina RTA v3.4.4 (RRID:SCR_014332), de-multiplexing and FASTQ file generation with Illumina bcl2fastq pipeline v. 2.20.0.422 (RRID:SCR_015058), producing ~480 million 300-bp read pairs, amounting to ~140 Gbp (Table 1).

## Read trimming and filtering

Raw, uncorrected Nanopore reads were used because correction can introduce artificial consensus sequences, and because we have additional, more accurate data (Illumina) available for correction. To ensure Illumina reads were as accurate as possible for the correction step, we used fastp version 0.20.0 (RRID:SCR_016962) [23] to filter and trim reads, with the following settings: discard read or pair if more than 3 Ns; require min length 130; require average quality 35. This reduced the number of read pairs to ~410 million (Table 1).

## Extraction and assembly of organelle-only Nanopore reads: round 1

To extract organelle-only reads from the full read sets, we used known sequences from related taxa as "baits". For the plastome, we used three coding sequences from *Acacia ligulata* (NC_026134.2) in FASTA nucleotide format. We chose the genes *rbcL*, *matK* and *ndhF* because these are all likely to be plastid-only genes and are also well conserved.



**Table 1.** Read and assembly statistics.

| Filename | No. seqs | No. base pairs | Minimum length (bp) | Maximum length (bp) |
|---|---|---|---|---|
| **Plastome – Illumina read statistics** | | | | |
| seq-reads/acacia_R1.fq.gz | 483,314,703 | 72,497,205,450 | 150 | 150 |
| seq-reads/acacia_R2.fq.gz | 483,314,703 | 72,497,205,450 | 150 | 150 |
| seq-reads/R1_fastp.fq.gz | 411,979,417 | 61,777,165,285 | 130 | 150 |
| seq-reads/R2_fastp.fq.gz | 411,979,417 | 61,777,336,578 | 130 | 150 |
| R1_extracted.fq.gz | 26,487,004 | 3,972,032,419 | 130 | 150 |
| R2_extracted.fq.gz | 26,487,004 | 3,972,036,119 | 130 | 150 |
| R1_extracted_subset.fq.gz | 133,367 | 20,000,084 | 130 | 150 |
| R2_extracted_subset.fq.gz | 133,367 | 19,999,906 | 130 | 150 |
| **Plastome – Nanopore read statistics** | | | | |
| seq-reads/acacia_promethion.fastq.gz | 5,468,251 | 57,567,959,340 | 3 | 171,656 |
| nano_extracted.fq.gz | 28,349 | 411,632,400 | 174 | 121,235 |
| nano_extracted_long.fq.gz | 901 | 40,013,219 | 36,575 | 121,235 |
| nano_extracted2.fq.gz | 70,080 | 1,025,810,636 | 6,290 | 121,235 |
| nano_extracted_long2.fq.gz | 864 | 40,007,899 | 38,997 | 121,235 |
| **Plastome – assembly statistics** | | | | |
| spades (via unicycler) | 12 | 135,939 | 128 | 62,974 |
| assembly_flye1.fasta | 2 | 172,743 | 80,753 | 91,990 |
| assembly_flye1_racon1.fasta | 2 | 173,100 | 80,836 | 92,264 |
| assembly_flye1_racon2.fasta | 2 | 173,087 | 80,815 | 92,272 |
| assembly_flye2.fasta | 2 | 172,914 | 80,744 | 92,170 |
| assembly_flye2_racon1.fasta | 2 | 173,284 | 80,806 | 92,478 |
| assembly_flye2_racon2.fasta | 2 | 173,253 | 80,794 | 92,459 |
| assembly_flye2_racon_pilon1.fasta | 2 | 173,693 | 80,945 | 92,748 |
| assembly_flye2_racon_pilon2.fasta | 2 | 173,695 | 80,947 | 92,748 |
| assembly_miniasm.fasta | 2 | 269,331 | 103,837 | 165,494 |
| assembly_miniasm_minipolished.fasta | 2 | 274,254 | 106,070 | 168,184 |
| assembly_miniasm_minipolished_pilon1.fa | 2 | 274,693 | 106,180 | 168,513 |
| assembly_raven.fasta | 1 | 209,875 | 209,875 | 209,875 |
| assembly_raven_pilon.fasta | 1 | 210,296 | 210,296 | 210,296 |
| assembly_unicycler.fasta | 1 | 173,902 | 173,902 | 173,902 |
| **Mitome – Illumina read statistics** | | | | |
| seq-reads/acacia_R1.fq.gz | 483,314,703 | 72,497,205,450 | 150 | 150 |
| seq-reads/acacia_R2.fq.gz | 483,314,703 | 72,497,205,450 | 150 | 150 |
| seq-reads/R1_fastp.fq.gz | 411,979,417 | 61,777,165,285 | 130 | 150 |
| seq-reads/R2_fastp.fq.gz | 411,979,417 | 61,777,336,578 | 130 | 150 |
| R1_extracted.fq.gz | 22,553,803 | 3,382,256,093 | 130 | 150 |
| R2_extracted.fq.gz | 22,553,803 | 3,382,255,114 | 130 | 150 |
| R1_extracted_subset.fq.gz | 666,830 | 100,000,063 | 130 | 150 |
| R2_extracted_subset.fq.gz | 666,830 | 99,999,574 | 130 | 150 |
| **Mitome – Nanopore read statistics** | | | | |
| seq-reads/acacia_promethion.fastq.gz | 5,468,251 | 57,567,959,340 | 3 | 171,656 |
| nano_extracted.fq.gz | 13,794 | 187,406,934 | 231 | 104,619 |
| nano_extracted_long.fq.gz | 13,794 | 187,406,934 | 231 | 104,619 |
| nano_extracted2.fq.gz | 14,098 | 215,256,727 | 6,686 | 104,619 |
| nano_extracted_long2.fq.gz | 12,327 | 200,007,409 | 9,284 | 104,619 |
| **Mitome – assembly statistics** | | | | |
| spades (via unicycler) | 161 | 823,167 | 127 | 55,079 |
| assembly_flye1.fasta | 4 | 870,672 | 18,983 | 497,136 |
| assembly_flye1_racon1.fasta | 4 | 869,346 | 18,785 | 496,324 |
| assembly_flye1_racon2.fasta | 4 | 868,229 | 18,674 | 495,712 |
| assembly_flye2.fasta | 6 | 905,290 | 42,974 | 391,756 |
| assembly_flye2_racon1.fasta | 6 | 905,216 | 42,670 | 391,835 |
| assembly_flye2_racon2.fasta | 6 | 904,454 | 42,544 | 391,296 |
| assembly_flye2_racon_pilon1.fasta | 6 | 905,790 | 42,651 | 391,825 |

| Table 1. (Continued) | | | | |
|---|---|---|---|---|
| Filename | No. seqs | No. base pairs | Minimum length (bp) | Maximum length (bp) |
| assembly_flye2_racon_pilon2.fasta | 6 | 905,783 | 42,648 | 391,827 |
| assembly_miniasm.fasta | 7 | 967,255 | 53,233 | 215,603 |
| assembly_miniasm_minipolished.fasta | 7 | 981,821 | 53,901 | 220,399 |
| assembly_miniasm_minipolished_pilon1.fa | 7 | 983,058 | 53,947 | 220,677 |
| assembly_raven.fasta | 7 | 933,304 | 65,567 | 220,209 |
| assembly_raven_pilon.fasta | 7 | 934,543 | 65,640 | 220,484 |
| assembly_unicycler.fasta | 4 | 818,342 | 54,008 | 578,782 |

The *rbcL* and *matK* genes are usually located at either end of the large single-copy (LSC) region, and *ndhF* is usually in the small single-copy (SSC) region; these are well spaced around the plastid so that long reads should be extracted with roughly even coverage. As the mitome is much larger than the plastome, we used all 38 of the coding sequences from the mitome of *Acacia ligulata* (NC_040998.1). We mapped the raw Nanopore reads (~5.5 million) to the baits with minimap2 version 2.17 (RRID:SCR_018550) [24] and used SAMTOOLS version 1.9 (RRID:SCR_002105) [25] to extract mapped reads. We then used Filtlong version 0.2.0 [26] to keep only the longest of the extracted reads up to a coverage of 250×, because assembly becomes more fragmented or impossible when coverage is too high (indeed, preliminary tests confirmed this with our data). For the plastome, we extracted ~28,000 reads, downsampled to 901 reads (longest ~121 kbp). For the mitome, we extracted ~14,000 reads, with no downsampling because coverage did not meet the cut-off (250×) (longest ~105 kbp) (Table 1). Extracted Nanopore reads were assembled with Flye version 2.8 [13] and the assembly was polished with two rounds of Racon version 1.4.11 (RRID:SCR_017642) [27].

### Extraction and assembly of organelle-only Nanopore reads: round 2

We used the first assembly as the bait file for the next round of extracting organelle reads from the original full read set. In Minimap2, we set a minimum match value to 5000, as preliminary tests showed that more leniency here resulted in too many reads being extracted to assemble properly. Again, we kept only the longest reads to a target coverage of 250×. From the ~5.5 million raw reads, for the plastome, we extracted ~70,000 reads (approximately twice as many as in round 1), downsampled to 864 reads (longest ~121,000 bp; the same as round 1). For the mitome, we extracted ~14,000 reads (similar to round 1), downsampled slightly to ~12,000 reads (longest ~105 kbp; same as round 1) (Table 1). As in the first round, these reads were then assembled with Flye and polished with two rounds of Racon. In testing, further rounds of Racon polishing made little difference.

### Extraction of organelle-only Illumina reads

Using the round 2 assembly as baits, we then extracted organelle-only reads from the filtered and trimmed Illumina reads (~410 million read pairs). The extracted read sets were then randomly downsampled to a coverage of 250× using Rasusa version 0.2.0 [28]. For the plastome, this resulted in ~26 million read pairs, downsampled to ~130,000 read pairs; for the mitome, this resulted in ~23 million read pairs, downsampled to ~670,000 read pairs (Table 1).

### Polishing the assembly with Illumina reads

The round 2 assembly was then polished with the extracted, downsampled Illumina reads, using two rounds of Pilon version 1.23 (RRID:SCR_014731) [29, 30], with a mindepth of 0.5 and fix set to bases (not contig breaks).

### Unicycler assembly

Using both the extracted Illumina and Nanopore reads, we used Unicycler version 0.4.8 [17] to perform a hybrid assembly. Unicycler first performs a short-read only assembly using Spades (RRID:SCR_000131) [31], and scaffolds this with long reads.

### Miniasm and Raven assemblies

Using the same read sets as in Unicycler (long and short reads), further polished long-read assemblies were made. Nanopore reads were assembled with Miniasm version 0.3_r179 [15] and Minipolish version 0.1.2 [16], and – separately – also with Raven version 1.1.10 (RRID:SCR_001937) [14]. Both assemblies were then further polished using Pilon [29, 30] with the Illumina reads.

### Assembly comparisons and verification

Assembly graphs were visualized with the Bandage tool GUI version 0.8.1 [32]. In particular, we used the BLAST v2.2.21 (RRID:SCR_001598) [33] tool within Bandage to compare assemblies. After loading a genome graph, a local BLAST database was built, and the query assembly file was compared; the assembly graph was then colored by BLAST hits. We made several comparisons: comparing each assembly to the Unicycler assembly, and comparing the Unicycler assembly to three closely related taxa from NCBI reference genomes. Further read-mapping was done to verify that long reads spanned multiple alternate structures.

### Scripts and computation

- Computation details: GNU/Linux OS, 16 CPUs, 32GB RAM, 3TB disk
- Custom script: assembler.sh, available at GitHub [34], with initial baits files, Illumina adapters, and a conda yaml file (with tools and versions)
- Plastome parameters: input genome size of 160,000 (size only has to be approximate) and target bases (for filtering) of 40 Mb (=coverage 250×)
- Mitome parameters: input genome size of 800,000 and target bases 200 Mb (=coverage 250×)

### Draft annotation

To annotate the Unicycler assemblies, we used the web service GeSeq v1.84 [19], which primarily uses BLAT (RRID:SCR_011919) for sequence comparison. As recommended [18], we used default settings, but enabled tRNAscan-SE and ARAGORN (RRID:SCR_015974). Settings for both organelles were: enable BLAT search [35] with protein search identity: 25; rRNA, tRNA, DNA search identity: 85; ARAGORN v1.2.38 [36] with "Allow overlaps" and "Fix introns" enabled, tRNAscan-SE v2.0.6 [37], and OGDRAW v1.3.1 (RRID:SCR_017337) [38] for visualization.

Specific settings for the plastome were: an additional HMMER (RRID:SCR_005305) profile search [39] enabled with chloroplast land plants, ARAGORN with genetic code for plant chloroplast, MPI-MP chloroplast references enabled. Specific settings for the mitome were:

| Table 2. Plastome – main assembly results. | | | | | |
|---|---|---|---|---|---|
| **Assembly type** | **Tool** | **Total length (bp)** | **Contigs (n)** | **Shortest contig (bp)** | **Longest contig (bp)** |
| Short-read | Spades | 135,939 | 12 | 128 | 62,974 |
| Long-read | Flye, polished | 173,695 | 2 | 80,947 | 92,748 |
| Hybrid | Unicycler | 173,902 | 1 | 173,902 | 173,902 |

| Table 3. Mitome – main assembly results. | | | | | |
|---|---|---|---|---|---|
| **Assembly type** | **Tool** | **Total length (bp)** | **Contigs (n)** | **Shortest contig (bp)** | **Longest contig (bp)** |
| Short-read | Spades | 823,167 | 161 | 127 | 55,079 |
| Long-read | Flye, polished | 905,783 | 6 | 42,648 | 391,827 |
| Hybrid | Unicycler | 818,342 | 4 | 54,008 | 578,782 |

ARAGORN genetic code standard; BLAT reference sequence NCBI RefSeq *Acacia ligulata* mitome NC_040998.1.

The annotations are summarized into output GenBank and GFF3 files. No additional manual editing or curation was performed, so these annotations act only as a first-pass overview of gene and feature content of the assemblies.

## RESULTS

### Overview

Plastome or mitome reads were extracted from the full read sets of Nanopore and Illumina data (Table 1). Short reads were assembled with Spades within Unicycler. Long reads were assembled with Flye, Miniasm and Raven, and assemblies were polished with long reads and then short reads. A hybrid assembly was performed with Unicycler. Assembly statistics are shown in Table 1, and summarized in Tables 2 and 3. Supplementary files available at Zenodo include assemblies, assembly graphs, and annotation files [40].

### Plastome short-read assembly

Despite being based only on short-reads, the plastome assembly is fairly well-resolved: there are 12 contigs, with the smallest contig being 128 bp in length, the largest contig being 62,974 bp, and the total length being 135,939 bp (Table 2, Figure 2). The assembly graph suggests the typical quadripartite structure of a long single-copy (LSC) region as the larger circle in the graph (see contigs labelled 1 and 2 in the figure, which are either side of some smaller unlabeled contigs), joined to inverted repeats (IRs) (labelled as contigs 3 and 4 in the figure, either side of some smaller unlabeled contigs) and a small single-copy region (SSC) (contig labelled 5 in the figure). The LSC has some unresolved repeats; this is because short reads (150 bp) cannot bridge these so are placed unambiguously in the assembly. The IR is a collapsed repeat of approximately twice the coverage. The total size is shorter than the expected ~160 kbp because the inverted repeat is only counted once.

### Plastome long-read assembly

As expected, this assembly is more fully resolved than the short-read assembly (Table 2, Figure 3). In the assembly graph, there are three contigs: the LSC, SSC, and an apparent collapsed repeat IR, separating the LSC and SSC. In the assembly file, these contigs have been joined into two contigs as such: one contig is ~81 kbp (IR + SSC + IR), and the other contig is ~92 kbp (LSC). This assembly was then polished with the long reads (using Racon)

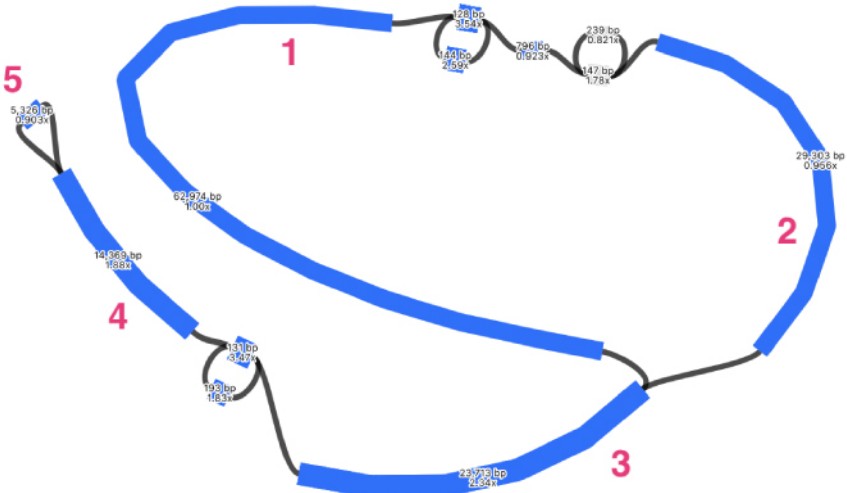

**Figure 2.** Assembly graph of the plastome based on short-read assembly with Spades, produced in the tool Bandage. Contigs are colored according to their match with the hybrid Unicycler plastome assembly, using the BLAST tool within Bandage. Labels show contig lengths and depths. Large numbers refer to particular contigs discussed in the text.

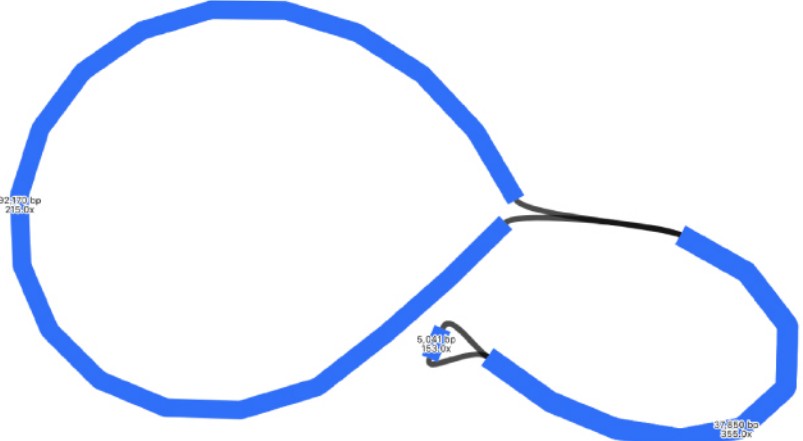

**Figure 3.** Assembly graph of the plastome based on a long-read assembly with Flye, produced in the tool Bandage. Contigs are colored according to their match with the hybrid Unicycler plastome assembly, using the BLAST tool within Bandage. Labels show contig lengths and depths. This graph is unpolished; contig sizes differ slightly after polishing with both Racon and Pilon.

and the short reads (using Pilon) which slightly increased the overall size by ~700 bp. The polished contig sizes are LSC (92,748 bp) and SSC joined by a collapsed IR (80,947 bp), total length: 173,695 bp (see Table 1 for all statistics).

## Plastome hybrid assembly

The hybrid assembly by Unicycler resolved the plastome assembly into a single circle, of length 173,902 bp, which is very similar to the long-read polished assembly size 173,695 bp (Table 2, Figure 4). As Unicycler is designed to work well for hybrid read sets like this, and

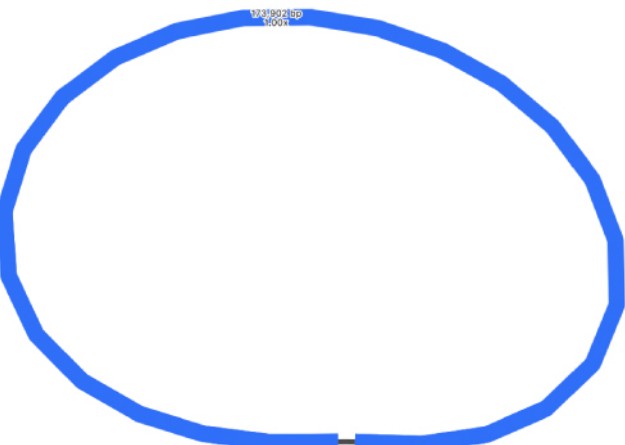

**Figure 4.** Assembly graph of the plastome based on a hybrid assembly with Unicycler, produced in the tool Bandage. Labels show contig length and depth.

can make use of short-read accuracy and long-read bridging, we consider this the best representation of the plastome in this analysis. Although this assembly is resolved into a circle, we keep in mind that there are probably two orientations of the SSC placement [7] and suspect that the long-read assembly alone does not call consensus on this ambiguity.

## Plastome assemblies with other tools

Plastomes were also assembled using two other tools. Long reads were assembled with Miniasm, then polished with Minipolish and Pilon, producing an assembly of ~275 kbp (Table 1, Figure 5). This assembly is much longer than both the Flye and Unicycler assemblies. Miniasm makes unitigs using the overlap-layout method, but with no consensus step. Here, because of either sequencing error and/or the multiple SSC orientations, it has probably assembled very similar regions that have not been collapsed into a consensus. BLAST reveals that this is probably the case, because almost the entire Miniasm assembly matches the Unicycler assembly (Figure 5). To better visualize the components of this assembly, we used BLAST to find locations of the LSC, SSC and IRs, taken from the Flye assembly in Figure 3 (Figure 6). Here, we can see that Miniasm has assembled reads into two contigs, one of which is almost the entire plastome, but that there is some ambiguity in the overlap of the other contig.

Long reads were also assembled with Raven, then polished with short reads and Pilon, producing a single contig of ~210 kbp (Table 1, Figure 7). Raven uses OLC in a slightly different way to Miniasm, and then includes a consensus step using Racon. Thus, because it is using OLC, this assembly is longer than the Flye/Unicycler assemblies, but because it includes a consensus step, it is shorter than the Miniasm assembly. Again, to better visualize the components of this assembly, we used BLAST to compare it to the LSC, SSC and IR regions taken from the Flye assembly in Figure 3 (Figure 8) where we can see that the IR has been assembled approximately three times, and the SSC twice.

## Plastome draft annotation

The draft annotation (Figure 9) is a visual first-pass approximation of the gene and feature content, rather than a highly polished finished annotation. Supplementary files, including

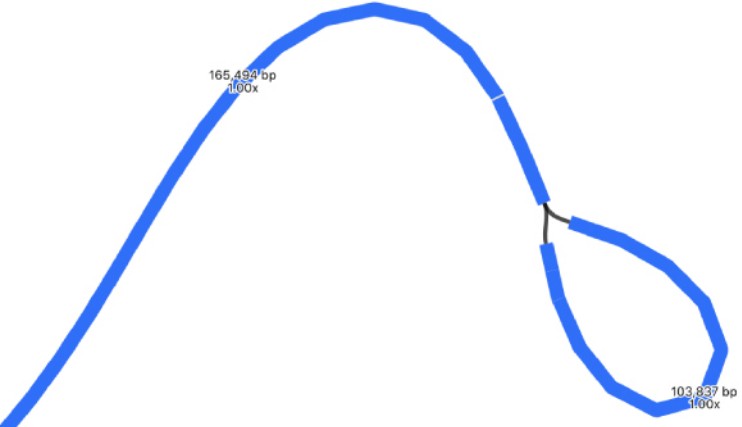

**Figure 5.** Assembly graph of the plastome based on a long-read assembly with Miniasm, produced in the tool Bandage. Contigs are colored according to their match with the hybrid Unicycler plastome assembly, using the BLAST tool within Bandage. The whole assembly is covered, showing that there is no new assembly here, only repeats of assembly sections that are present in the Unicycler assembly. Labels show contig lengths and depths. This graph is unpolished; contig sizes differ slightly after polishing with Pilon.

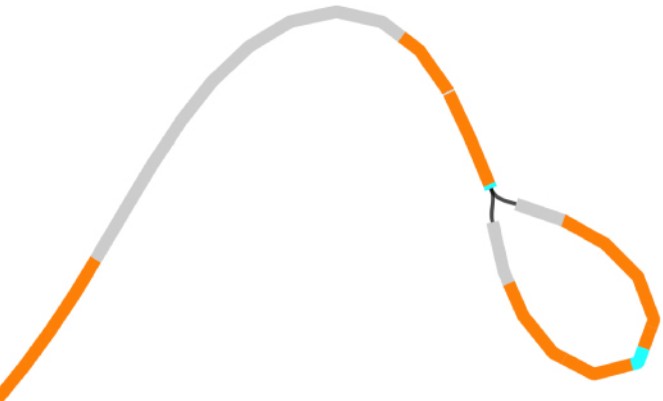

**Figure 6.** Assembly graph of the plastome based on a long-read assembly with Miniasm, produced in the tool Bandage. Contigs are colored according to their match with particular sections of the Flye assembly: regions in gray match the LSC, regions in blue match the SSC, and regions in orange match the IR.

GenBank and GFF3 formats of this annotation are available for researchers to further explore this annotation [40].

## Plastome summary

Based on the Unicycler assembly here, and annotations by GeSeq, the assembly of the *Acacia pycnantha* plastome is broadly similar to previous results found in other *Acacia* species (Table 4). As an additional visual comparison, we used the BLAST tool within Bandage to compare the Unicycler assembly of *A. pycnantha* with plastomes of related species in subfamily Caesalpinioideae: *A. ligulata* (NCBI Reference Sequence NC_026134.2), *Leucaena trichandra* (NCBI Reference Sequence NC_028733.1), and *Haematoxylum brasiletto* (GenBank KJ468097.1). There are no large unmatched sections in the *A. pycnantha* assembly that would indicate potentially novel regions or misassembly (Figure 10).



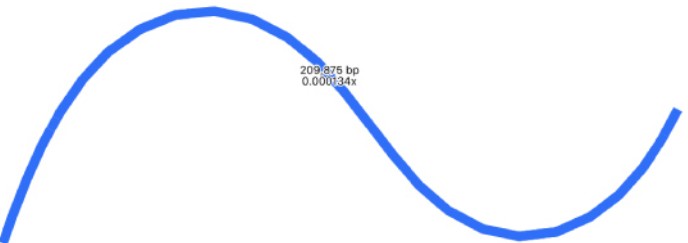

**Figure 7.** Assembly graph of the plastome based on a long-read assembly with Raven, produced in the tool Bandage. Contigs are colored according to their match with the hybrid Unicycler plastome assembly, using the BLAST tool within Bandage. The whole assembly is covered showing that there is no new assembly here, only repeats of assembly sections that are present in the Unicycler assembly. Labels show contig lengths and depths. This graph is unpolished; contig sizes differ slightly after polishing with Pilon.

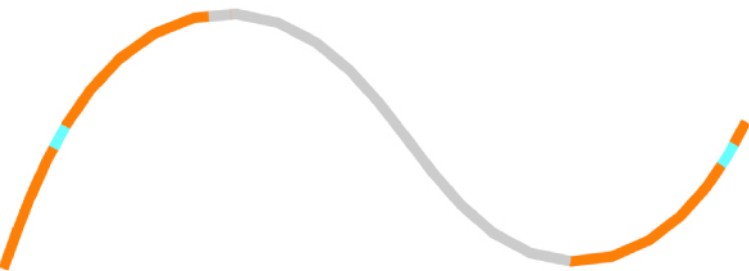

**Figure 8.** Assembly graph of the plastome based on a long-read assembly with Raven, produced in the tool Bandage. Contigs are colored according to their match with particular sections of the Flye assembly: regions in gray match the LSC, regions in blue match the SSC, and regions in orange match the IR.

**Table 4.** Comparisons between different *Acacia* plastome assemblies, showing number of base pairs in different genome components.

| Species | Total length (bp) | Length of LSC (bp) | Length of IR (bp) | Length of SSC (bp) |
|---|---|---|---|---|
| *Acacia pycnantha* | 173,902 | 92,772 | 38,028 | 5071 |
| *Acacia ligulata* | 174,233 | 92,798 | 38,225 | 4985 |
| *Acacia dealbata* | 174,217 | 92,753 | 38,254 | 4956 |

*Acacia ligulata* statistics are from [41]; *Acacia dealbata* statistics are from [42]. *Acacia pycnantha* statistics are derived from the GeSeq annotation, visualized in OGDRAW.

## Mitome short-read assembly

The mitome assembly based on short reads has 161 contigs, ranging from 127 bp to ~55,000 bp in length, giving a total length of 823,167 bp (Table 3). As expected, the assembly graph shows some unresolved ambiguity, at least one dead end, and several very small fragments (Figure 11).

## Mitome long-read assembly

The long-read assembly of the mitome is a vast improvement over the short-read assembly (Table 3) in terms of contig lengths and contiguity. The number of contigs has reduced from 161 to 6, the shortest contig has increased in size from 127 bp to ~43 kbp, and the longest has increased from ~55 kbp to ~392 kbp. Total length has increased from ~823 kbp to ~906 kbp. The assembly graph is much less tangled: there are two possibly circular segments of ~93 kbp and ~108 kbp, and the remainder forms a single structure, albeit with



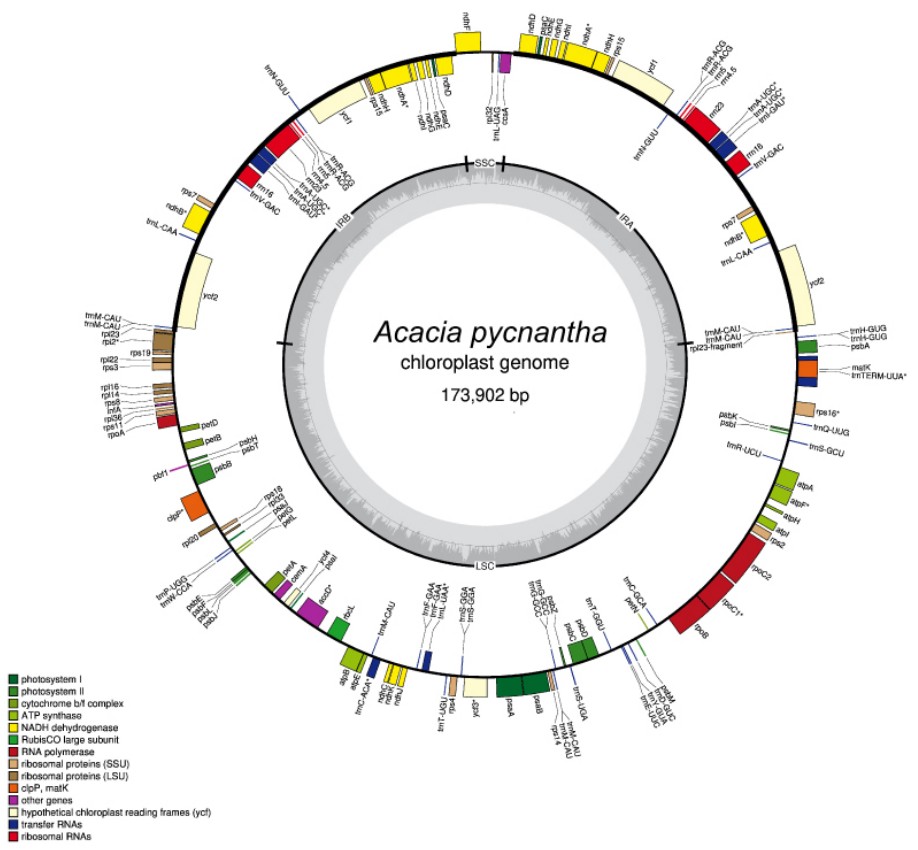

**Figure 9.** Annotated plastome of *Acacia pycnantha*, based on Unicycler assembly, and produced by the GeSeq tool and OGDRAW.

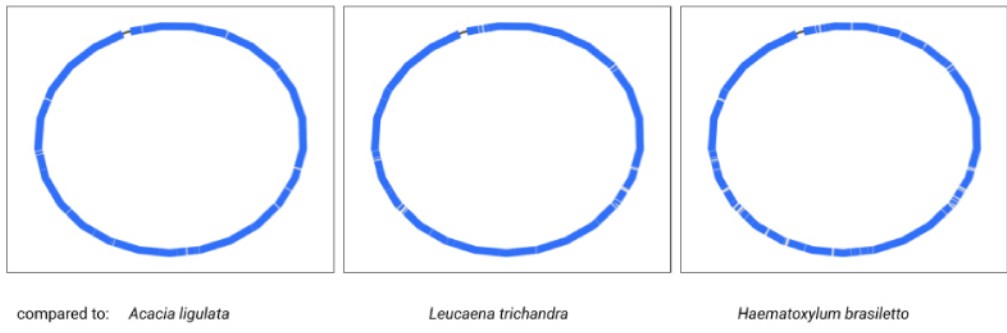

compared to:    *Acacia ligulata*        *Leucaena trichandra*        *Haematoxylum brasiletto*

**Figure 10.** Comparison of the *Acacia pycnantha* Unicycler assembly with plastomes from related species. The contig is colored according to its match with these assemblies, using the BLAST tool within Bandage. No novel regions or misassembly are evident within *A. pycnantha*.

some ambiguous regions (Figure 12). A note that contigs in the assembly graph are different to the contigs in the FASTA file: FASTA file contigs include only the longest unambiguous paths and so are broken at repeats.

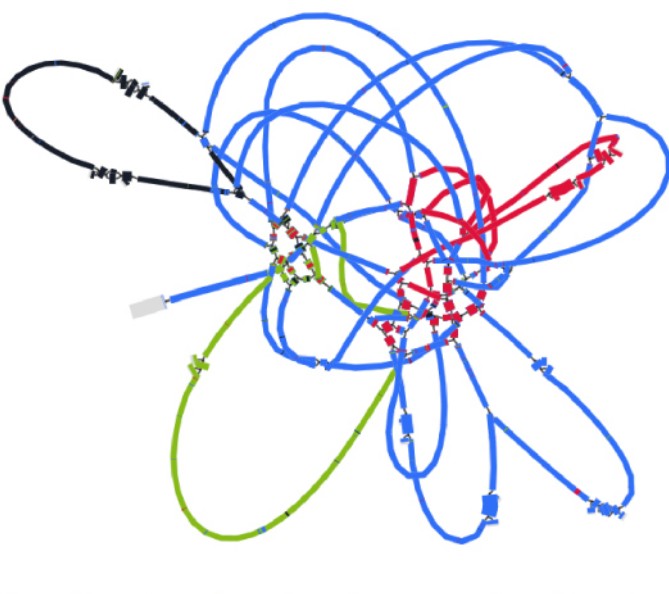

**Figure 11.** Assembly graph of the mitome based on short-read assembly with Spades, produced in the tool Bandage. Contigs are colored according to their match with the hybrid Unicycler mitome assembly, using the BLAST tool within Bandage.

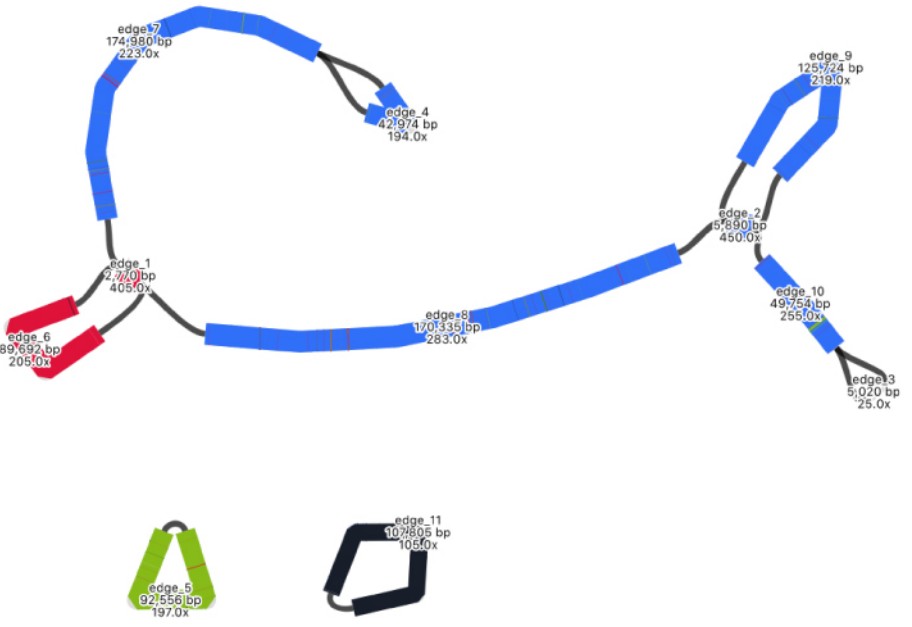

**Figure 12.** Assembly graph of the mitome based on long-read assembly with Flye, produced in the tool Bandage. Contigs are colored according to their match with the hybrid Unicycler mitome assembly, using the BLAST tool within Bandage. Labels show contig lengths and depths. This graph is unpolished; contig sizes differ slightly after polishing with both Racon and Pilon.

## Mitome hybrid assembly

The mitome hybrid assembly produced by Unicycler is well resolved, with some apparent improvements over the long-read assembly by Flye (Table 3, Figure 13). The number of

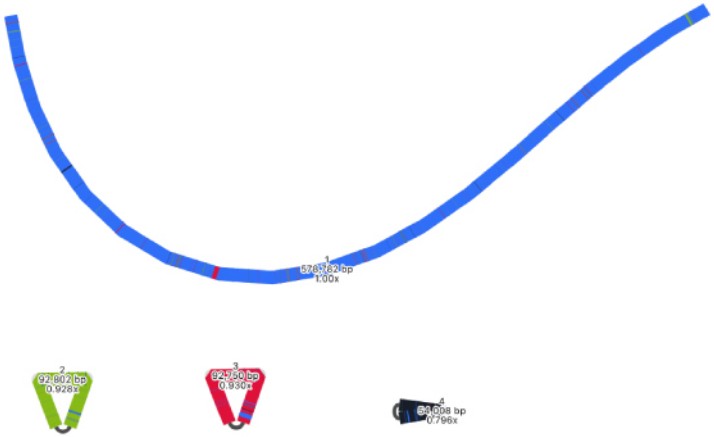

**Figure 13.** Assembly graph of the mitome based on a hybrid assembly with Unicycler, produced in the tool Bandage. Labels show contig names, length and depth. To compare this assembly with the other assemblies in this analysis, contigs have been colored according to their match with self, using the BLAST tool within Bandage.

contigs has decreased from 6 to 4, the shortest contig has increased in size from ~43 kbp to ~54 kbp, and the longest contig from ~392 kbp to ~579 kbp. Total size has decreased from ~906 kbp to ~818 kbp. There are three apparent circular segments of sizes ~93 kbp, ~93 kbp and ~54 kbp.

As with the plastome assembly, based on the strengths of Unicycler in working with hybrid read sets, we consider this Unicycler assembly the best representation of the mitome in this analysis. However, by also considering the long-read Flye assembly, we can explore the complexity of this genome structure further. The Flye assembly joins a longer section together, indicating how a particular segment (shown in red) may be integrated. Although the Flye assembly is substantially longer than the Unicycler assembly, a BLAST comparison (Figure 12) shows that all components match well to the Unicycler assembly, indicating that additional length may be from a similar repeat region that has not been collapsed. The Flye assembly also shows that one of its circular segments (shown in black) is twice the size of the similar segment in the Unicycler assembly, which again suggests a repeat region that has not been collapsed. In the Unicycler assembly, BLAST shows that this segment matches to a related *Acacia* species (Figure 17). However, whether these repeat regions are truly independent and should be collapsed is unclear, demonstrating the complexity of this structure.

## Mitome assemblies with other tools

Mitomes were also assembled with two other tools. Long reads were assembled with Miniasm, then polished with Minipolish and Pilon, producing an assembly of ~983 kbp (Table 1, Figure 14). As with the plastome results, this assembly is much longer than the Flye and Unicycler assemblies, which is expected as Miniasm has no consensus step. Using BLAST to compare this assembly with that produced by Unicycler, all sections match the Unicycler assembly (Figure 14).

Long reads were also assembled with Raven, then polished with short reads and Pilon, producing an assembly of ~935 kbp (Table 1, Figure 15). As with the plastome results, this Raven assembly is shorter than the Miniasm assembly, but is longer than the Flye/Unicycler

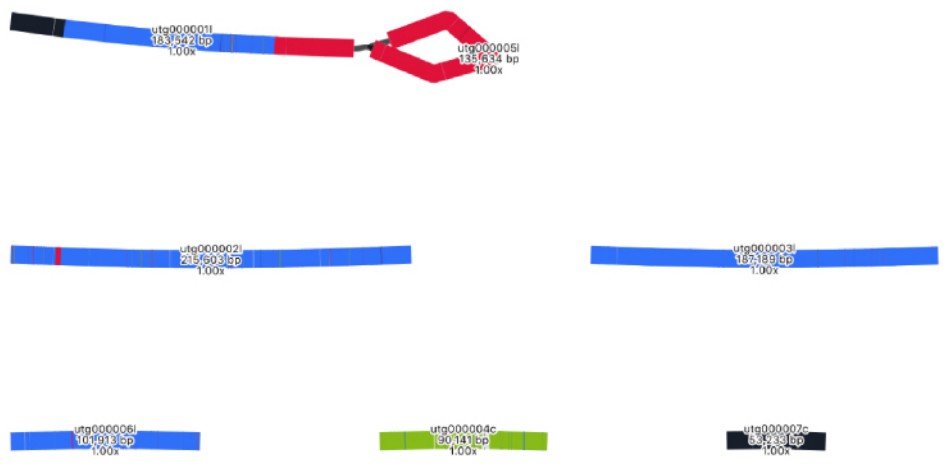

**Figure 14.** Assembly graph of the mitome based on a long-read assembly with Miniasm, produced in the tool Bandage. Contigs are colored according to their match with the hybrid Unicycler mitome assembly, using the BLAST tool within Bandage. The whole assembly is covered showing that there is no new assembly here, only repeats of assembly sections that are present in the Unicycler assembly. Labels show contig names, lengths and depths. This graph is unpolished; contig sizes differ slightly after polishing with Pilon.

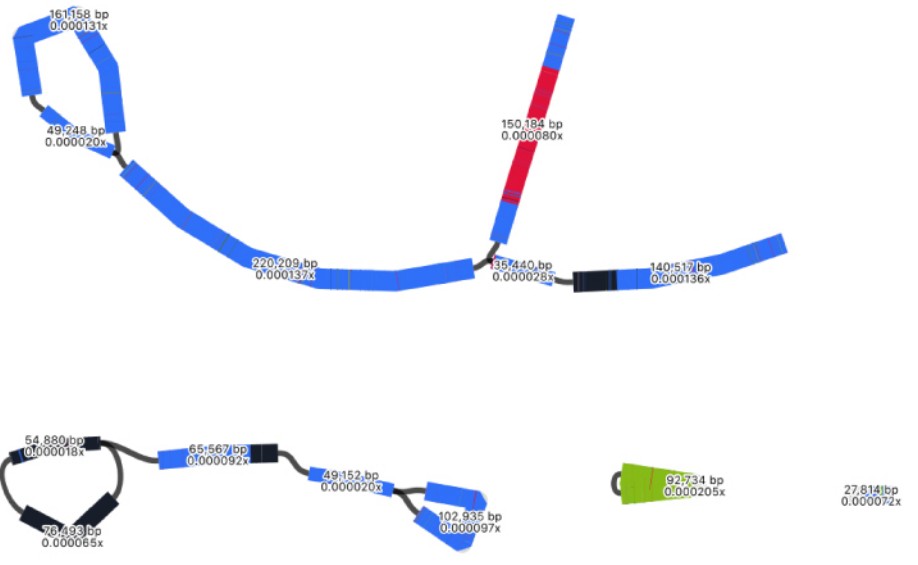

**Figure 15.** Assembly graph of the mitome based on a long-read assembly with Raven, produced in the tool Bandage. Contigs are colored according to their match with the hybrid Unicycler mitome assembly, using the BLAST tool within Bandage. The whole assembly is covered showing that there is no new assembly here, only repeats of assembly sections that are present in the Unicycler assembly. Labels show contig lengths and depths. This graph is unpolished; contig sizes differ slightly after polishing with Pilon.

assemblies because of the algorithm employed. A BLAST comparison with the Unicycler assembly confirms that all sections match (Figure 15).

An interesting result from the Minisam and Raven assemblies is the placement of a particular segment, colored in black. In the Unicycler assembly, this segment is circularized, but these assemblies indicate how the segment may join to other parts of the genome (Figures 14, 15).

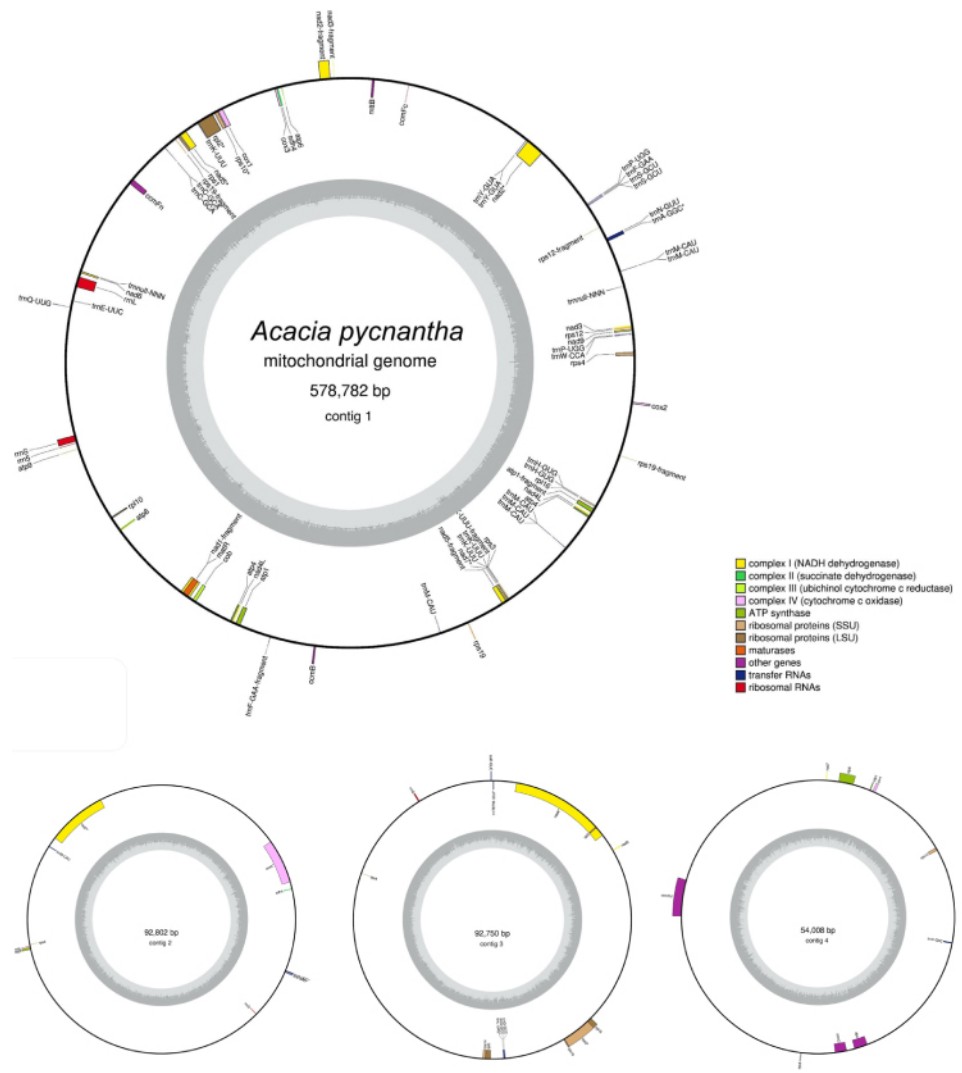

**Figure 16.** Annotated mitome of *Acacia pycnantha*, based on Unicycler assembly, and produced by the GeSeq tool and OGDRAW.

## Mitome draft annotation

The draft annotation (Figure 16) is presented to provide a first-pass visualization of gene and feature content. To further explore this annotation, supplementary files available at Zenodo [40] include GenBank and GFF3 formats of this annotation.

## Mitome summary

Based on the Unicycler assembly, the assembly of the *Acacia pycnantha* mitome is 818,342 bp in length, and may be arranged in a long linear piece and three smaller circular segments. Alternative assembly results suggest that some circular segments can be incorporated into the larger structure. In comparison, the closest sequenced relative, *Acacia ligulata* is substantially shorter, with a total length of 698,138 bp [43]. This was assembled with short Illumina reads into 10 contigs, followed by manual editing and joining.

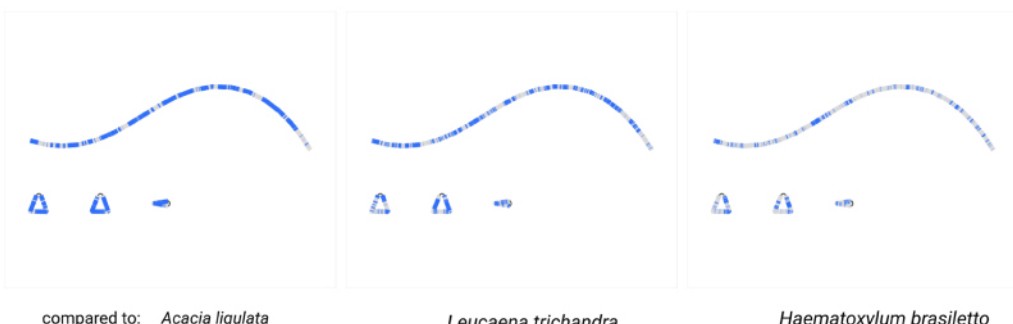

compared to:   *Acacia ligulata*          *Leucaena trichandra*          *Haematoxylum brasiletto*

**Figure 17.**   Comparison of the *Acacia pycnantha* Unicycler assembly with mitomes from related species. The contig is colored according to its match with these assemblies, using the BLAST tool within Bandage. Increasing difference is seen in concert with increasing phylogenetic distance, from left to right.

Interestingly, the work on *Acacia ligulata* suggested the possible existence of alternative structures in the form of head-to-tail concatemers, which is consistent with the alternate forms assembled for *Acacia pycnantha* mitomes herein.

As an additional visual comparison, we used the BLAST tool within Bandage to compare the Unicycler assembly of *Acacia pycnantha* with mitomes of related species in subfamily Caesalpinioideae: *Acacia ligulata* (NCBI Reference Sequence NC_040998.1), *Leucaena trichandra* (NCBI Reference Sequence NC_039738.1), and *Haematoxylum brasiletto* (NCBI Reference Sequence NC_045040.1). In contrast to the plastome assembly comparisons, there is much non-matching sequence in the *Acacia pycnantha* assemblies, increasing in concert with phylogenetic distance (Figure 17).

## Mitome structural variation

A major aim of this study is to understand more about multiple structures of an organelle genome that may exist simultaneously. Although various assembly results suggest this, we performed an extra manual check to deduce whether long reads would span multiple structures. To do this, we specifically looked at the location of the red contig, with a size of ~90 kbp. In the long-read Flye assembly (Figure 12), this contig (=edge 6) is integrated into the large blue connected component, and located between the repeat region of 2,770 bp (=edge 1). However, in the Unicycler assembly (Figure 13), the red contig is excised as an independent circular contig, and a small fragment is also present in the long linear contig.

The question is: can both these structures exist? Do long reads support both configurations? The two paths to compare in Figure 12 are a structure that includes edge 6 (edge 8–edge 1–edge 6–edge 1–edge 7) and a structure that excludes edge 6 (edge 8–edge 1–edge 7). We mapped the long reads to these paths to visually inspect whether there was support for both assemblies. To do this, we drew the Flye assembly in Bandage in double mode (Figure 18), extracted nodes in the two paths of interest (keeping direction consistent), reduced the lengths of the outer contigs (edges 7 and 8) to only 10,000 bp, and combined them into a single FASTA file of paths. Then, we used this FASTA file as bait to extract matching reads from the full mitome Nanopore read set with minimap2 [24], and visualized the bam track (available at Zenodo [40]) in JBrowse [44] in Galaxy [45]. This confirmed that long reads span both paths and thus both structural configurations, where the ~90 kbp red

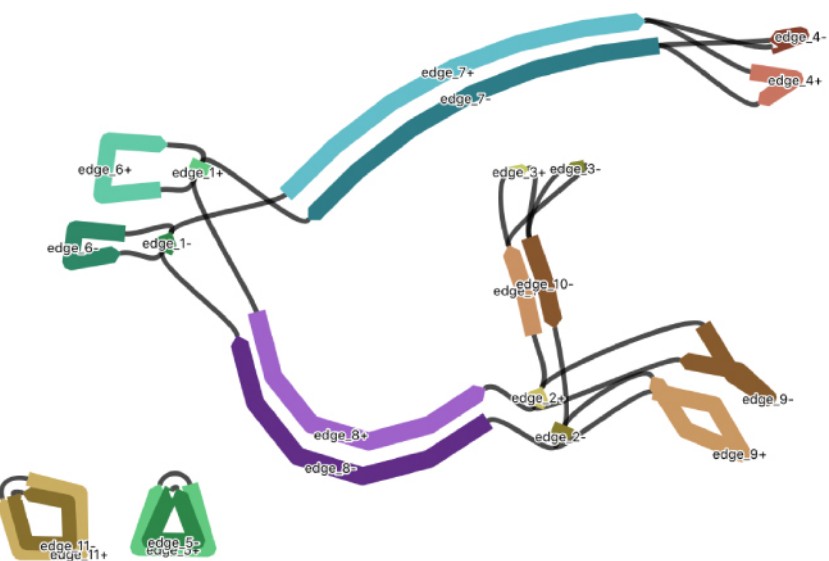

**Figure 18.** Tracing alternate paths through the mitome assembly graph, to include or exclude edge 6. Graph is drawn in Bandage in double mode, so the path directionality is maintained when nodes are extracted.

contig can be incorporated into the larger mitome structure (Figure 12) or not (Figure 13).

## DISCUSSION

Long sequencing reads are becoming the "new normal" for genome assembly projects, providing new ways to investigate structural complexity. In this work, we successfully extracted organelle-only reads from full nuclear + organelle read sets, and assembled the reads under various algorithms in well-tested tools. In this case, we consider that the hybrid short- and long-read assembly produced by Unicycler gave the best representations of the *Acacia pycnantha* mitome and plastome. Draft annotations have been presented from the GeSeq tool.

Additional assemblies have suggested the existence of multiple mitome configurations; a hypothesis supported by long reads that span alternate assembly graph paths. This builds on an increasing body of work that refutes the existence of a single, static, circular mitochondrial genome [3, 4, 46, 47].

There are many avenues to explore to improve both assemblies and annotations, so we consider the assemblies presented here as "version 1". New technologies, such as PacBio HiFi sequencing, are improving long-read fidelity. Oxford Nanopore raw sequencing data can benefit from being re-basecalled with new tools [48], and trained on relevant taxonomic data [49]. Long-read specific assemblers are continually optimized, particularly to error profiles, and there is a large focus on improving the assembly of repeat regions [50].

One option to explore in further research is that multiple structures may be better assembled via metagenomic approaches. Alternate structures could be thought of as part of a metagenomic pool, and reads clustered and assembled accordingly, considering that abundances of alternate forms may not be equal.

In either case, the increased use of long reads and *de novo* assembly will further improve organelle assemblies and pave the way for fuller genomic comparison across species.

## AVAILABILITY OF SOURCE CODE AND REQUIREMENTS

A copy of the assembly script (assembler.sh) is available in Zenodo [34] and in GitHub [40], with instructions on how to run the script and the required inputs and tools. The latest release (v1.1) contains the MIT license.

## DATA AVAILABILITY

Raw sequence data has been deposited in NCBI under BioProject PRJNA752212. Extracted organelle data is available at Zenodo [51]. Supplementary files also available at Zenodo [40] include, for each organelle genome: 14 assemblies in fasta format, and associated GFA format file if available (not all stages produce this file), as well as the Spades GFA from Unicycler. For each Unicycler assembly there is a set of annotation files that include GenBank and GFF3 formats, and outputs from HMMER, ARAGORN, and tRNAscan-SE. There is also a bam file of reads mapped to alternate assembly paths for the mitome to explore the 90 kbp contig of interest.

## DECLARATIONS
## LIST OF ABBREVIATIONS

bp: base pair; Gbp: gigabase pair; IR: inverted repeat; kbp: kilobase pair; LSC: large single copy; NCBI: National Center for Biotechnology Information; OLC: overlap–layout–consensus; PacBio: Pacific Biosciences; SSC: small single copy.

## ETHICAL APPROVAL

Not applicable.

## COMPETING INTERESTS

The authors declare that they have no competing interests.

## FUNDING

This project was funded by the Genomics for Australian Plants Framework Initiative consortium, which is supported by funding from Bioplatforms Australia (enabled by the Australian National Collaborative Research Infrastructure Strategy), the Ian Potter Foundation, Royal Botanic Gardens Victoria, the Royal Botanic Gardens and Domain Trust, Commonwealth Scientific and Industrial Research Organisation (CSIRO), Centre for Australian National Biodiversity Research and the Department of Biodiversity, Conservation and Attractions, Western Australia.

## AUTHORS' CONTRIBUTIONS

Funding acquisition: D.C., D.M.; Project administration: F.U., D.M.; Data collection: T.M.; Conceptualization and Software: A.S.; Formal analysis: A.S., T.M.; Writing original draft: A.S.; Review and editing: all authors.

## ACKNOWLEDGEMENTS

We would like to acknowledge the contribution of the Genomics for Australian Plants Framework Initiative consortium in the generation of data used in this publication.



We would like to thank the Australian National Botanic Gardens for access to plant material, in particular Tamera Beath; and Theo Allnutt for uploading the Acacia project data to NCBI. Anna Syme wishes to thank Torsten Seemann for discussion on the iterative bait + assembly approach, and Shaun Jackman for discussion on mitochondrial genome structure and for suggesting the utility of the BLAST v2.2.21 tool within the Bandage v0.8.1 assembly viewer.

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
