## [Reviewer Report]

Reviewer name and names of any other individual's who aided in reviewerAboozar SoorniDo you understand and agree to our policy of having open and named reviews, and having your review included with the published manuscript. (If no, please inform the editor that you cannot review this manuscript.)YesIs the language of sufficient quality?YesPlease add additional comments on language quality to clarify if neededIs there a clear statement of need explaining what problems the software is designed to solve and who the target audience is? NoAdditional CommentsIs the source code available, and has an appropriate Open Source Initiative license <a href="https://opensource.org/licenses" target="_blank">(https://opensource.org/licenses)</a> been assigned to the code?YesAdditional CommentsAs Open Source Software are there guidelines on how to contribute, report issues or seek support on the code?YesAdditional CommentsIs the code executable?YesAdditional CommentsIs installation/deployment sufficiently outlined in the paper and documentation, and does it proceed as outlined?YesAdditional CommentsIs the documentation provided clear and user friendly?YesAdditional CommentsIs there a clearly-stated list of dependencies, and is the core functionality of the software documented to a satisfactory level?YesAdditional CommentsHave any claims of performance been sufficiently tested and compared to other commonly-used packages? NoAdditional CommentsAre there (ideally real world) examples demonstrating use of the software? NoAdditional CommentsIs automated testing used or are there manual steps described so that the functionality of the software can be verified?NoAdditional CommentsAny Additional Overall Comments to the AuthorRecommendationAccept

---

## [Reviewer Report]

Reviewer name and names of any other individual's who aided in reviewerWeiwen WangDo you understand and agree to our policy of having open and named reviews, and having your review included with the published manuscript. (If no, please inform the editor that you cannot review this manuscript.)YesIs the language of sufficient quality?YesPlease add additional comments on language quality to clarify if neededIs there a clear statement of need explaining what problems the software is designed to solve and who the target audience is? YesAdditional CommentsIs the source code available, and has an appropriate Open Source Initiative license <a href="https://opensource.org/licenses" target="_blank">(https://opensource.org/licenses)</a> been assigned to the code?YesAdditional CommentsAs Open Source Software are there guidelines on how to contribute, report issues or seek support on the code?YesAdditional CommentsIs the code executable?YesAdditional CommentsIs installation/deployment sufficiently outlined in the paper and documentation, and does it proceed as outlined?YesAdditional CommentsIs the documentation provided clear and user friendly?YesAdditional CommentsIs there a clearly-stated list of dependencies, and is the core functionality of the software documented to a satisfactory level?YesAdditional CommentsHave any claims of performance been sufficiently tested and compared to other commonly-used packages? YesAdditional CommentsAre there (ideally real world) examples demonstrating use of the software? YesAdditional CommentsIs automated testing used or are there manual steps described so that the functionality of the software can be verified?NoAdditional CommentsI suggest the authors provide a test sample on the Github.Any Additional Overall Comments to the AuthorThis manuscript is easy to read, with good writing and details of methods. By comparing the long-read, short-read and hybrid assembly, this manuscript found out the best approach to assemble plastid and mitochondrial genome. Additionally, authors considered the multiple structures of plastid and mitochondrial genome, and providing a new and carefully designed method to assemble and assess the complex mitochondrial genome. 
While this manuscript represents a solid work and it was interesting to read it, I have some minor concerns which should be fixed to improve the quality of the manuscript.

Line 240-247: Maybe it could be easier to understand if authors can number each contig. For example, “The assembly graph suggests the typical quadripartite structure of a LSC (contig 1-7) as the larger circle in the graph…...”. In some figures, authors numbered the contigs, but in some did not. 
Also, in the figure 2, why does the SSC region also have almost 2x coverage (1.88x)?

Line 253-258: In the figure 3, it is three contigs (92 kb, 38 kb and 5kb) rather than two contigs (81 kb and 92 kb) that this manuscript described. I guess authors put the wrong figure?

Line 283-285: It is a smart method to clearly show the assembly details.

Line 385-387: Does it mean that the black segment (edge 11) in Figure 12 is consisted of two highly similar (or the same) regions? Have authors tried to do a simple BLAST to confirm this?
RecommendationMinor Revisions